# Improving Properties of an Experimental Universal Adhesive by Adding a Multifunctional Dendrimer (G-IEMA): Bond Strength and Nanoleakage Evaluation

**DOI:** 10.3390/polym14071462

**Published:** 2022-04-03

**Authors:** Joana Vasconcelos e Cruz, António H. S. Delgado, Samuel Félix, José Brito, Luísa Gonçalves, Mário Polido

**Affiliations:** 1Instituto Universitário Egas Moniz (IUEM), Campus Universitário, Quinta da Granja, Monte de Caparica, 2829-511 Almada, Portugal; aldelgado@egasmoniz.edu.pt (A.H.S.D.); samuelfelixmem@hotmail.com (S.F.); jbrito@egasmoniz.edu.pt (J.B.); lmlgoncalves@gmail.com (L.G.); mpolido@egasmoniz.edu.pt (M.P.); 2Centro de Investigação Interdisciplinar Egas Moniz (CiiEM), Campus Universitário, Quinta da Granja, Monte de Caparica, 2829-511 Almada, Portugal; 3Division of Biomaterials & Tissue Engineering, UCL Eastman Dental Institute, University College London, London NW3 2PF, UK

**Keywords:** bisphenol A, BPA-free, dendrimers, dental adhesives, dentin bonding, G-IEMA, nanoleakage

## Abstract

A vast number of adhesive formulations exist currently. However, available adhesives still have several drawbacks such as increased hydrophilicity, polymerization deficiency, potential cytotoxicity and limited monomer interdiffusion within dentin. To improve material properties, a Bisphenol A-free adhesive containing a novel dendrimer G(2)-isocyanatoethyl methacrylate (G-IEMA) in replacement of Bis-GMA was made and tested. Sound human molars were sectioned to expose mid-coronal dentin, which was bonded using four adhesives—Futurabond, Scotchbond Universal and experimentals EM1 and EM2. The experimental adhesive EM2 contained G-IEMA, while EM1 had Bis-GMA. Groups were further allocated to two different adhesive strategies: etch-and-rinse (20 s etching) or self-etch. Immediate (24 h) microtensile bond strength to dentin (*n* = 5) was tested using a universal testing machine (1 mm/min, 5 kN; Shimadzu AGS-X Autograph, Tokyo, Japan), while the ultrastructure of the interface (*n* = 2) was assessed using scanning electron microscopy-energy dispersive spectroscopy (SEM-EDS). Nanoleakage expression was evaluated using silver nitrate penetration and posterior SEM-EDS analysis (*n* = 3). Linear mixed models/Generalized models were used for inferential statistics (5% significance level). Bond strength results did not depend on the adhesive choice, although differences were found between strategies (*p* < 0.001). Regarding nanoleakage, when applied in an etch-and-rinse mode, experimental adhesives scored lower nanoleakage means than Futurabond and Scotchbond Universal. The novel adhesive shows interesting interfacial properties, with favorable nanoleakage results and a bond strength to dentin that matches current commercial adhesives.

## 1. Introduction

Bonding strategies in restorative dentistry achieved clinically acceptable longevity by securing different types of adhesion mechanisms (i.e., mechanical or chemical) to enamel and dentin, due to their dissimilar nature and composition [1,2]. Adhesion to dentin, albeit complex, relies on the formation of a hybrid layer (HL), a true bioengineered complex with intrinsic properties [3,4]. This complex is an interdiffusion zone, constructed from elements of the adhesive, type I dentinal collagen, residual hydroxyapatite, and debris [5]. Despite being the subject of extensive research, many factors contribute to the adhesive interface in dentin still being dubbed the weakest link [6,7]. On one hand, the organic content and the significant amount of water in the dentinal structure increases the difficulty in resin interdiffusion and may provoke degradation phenomena, leading to a less predictable bonding outcome. On the other hand, the hydrophilicity and acidity of the adhesive components are also known to contribute to the instability of the bonding interface. To overcome these limitations, several formulations and new adhesive strategies have been studied and proposed in the literature in recent years [8,9].

To this end, universal adhesives (UA) were introduced in the market more than 10 years ago as a novel, versatile material that can be applied in three distinct adhesive strategies, depending on the use of separate acid, and if this acid is used only in enamel or both enamel/dentin [10]. Currently, there is a vast array of commercial brands marketing universal adhesives, containing different functional monomers and thus different chemical compositions. Such differences have been proposed to overcome the drawbacks mentioned above, such as hydrophilicity, limited interdiffusion of resin components and degradation over time, that all contribute to the inevitable failure of the restoration [11,12]. Additionally, nanoleakage phenomena within the interdiffusion zone is still considered the main reason for the adhesive failures, owing to silent hydrolysis mechanisms interrelated to the matrix-bound enzymatic degradation at the resin-dentin interface [12,13].

An example of a strategy to overcome these limitations is the addition of innovative cross-linking monomers that can confer desired hydrophobicity, while still improving the degree of monomer conversion and mechanical properties. Consequently, these physicochemical parameters help to lower the amount of water within the HL, bound to degrade it, while also guaranteeing intimate interaction between the adhesive and the collagen fibers within the demineralized dentin matrix [14].

Recently, Vasconcelos e Cruz et al. (2019 & 2020) introduced a methacrylate dendrimer, used as a monomer in dental adhesive co-polymers. G-IEMA is a second generation (following reaction product and purification) dendrimer, with a star shape containing eight methacrylate extremities, which according to Yu et al. (2014) [15] play a role in increasing conversion rates and reducing associated volumetric shrinkage when incorporated into resin composites. Its use in dental adhesives, in replacement of Bis-GMA, has shown very interesting results regarding physicochemical and mechanical properties. It also significantly improved the conversion rates, reduced the polymerization shrinkage [16] and contributed to an increase in immediate dentin bond strength when an etch-and-rinse protocol was used. However, its ultra-mild pH could have harmed the efficacy of the self-etch approach. In addition, the behavior of such formulations after an aging period is unknown [17].

Considering all these aspects, re-formulation of a dental adhesive containing G-IEMA is advocated to overcome the identified limitations. Thus, it is important to evaluate dentin interfaces regarding immediate bonding and resulting nanoleakage of such experimental adhesives, as this area remains unstudied. This investigation could support the use of G-IEMA as an alternative to the popular Bis-GMA in future adhesive formulations. In fact, this dendrimer could theoretically improve the degree of conversion in situ, improving the quality of the bond by enhancing the cohesive properties of the hybrid layer. This can then protect the interface, thereby improving longevity.

The aim of this study was thus to investigate the use of dendrimer G-IEMA, creating a BPA-free formulation, on the immediate dentin bond strength and its impact on nanoleakage after a 3 month aging period. The null hypotheses were that: there were no differences between the experimental and commercial adhesives tested, or strategies under study (etch-and-rinse and self-etch) (1) in immediate microtensile bond strength and quality of the adhesive interface formed and (2) in nanoleakage after 3 months of aging.

## 2. Materials and Methods

### 2.1. Formulation of the Experimental Adhesives

All reagents and solvents used for the formulations and described in Table 1 were of analytical grade and purchased from Merck KGaA/Sigma-Aldrich (Darmstadt, Germany), with the exception of 10-MDP (10-methacryloyloxy decyldihydrogen phosphate) which was acquired from Watson International Ltd. (Kunshan, China).

The chemical composition of the experimental adhesives (EM1 and EM2) was defined based on the composition of commercial adhesives used as control groups, which were SBU (Scotchbond Universal, 3M ESPE, Saint Paul, MN, USA) and FUT (Futurabond, VOCO GmbH, Cuxhaven, Germany) and also considering the composition of the similar experimental adhesives tested in previous studies [17]. For the formulation of EM2, G-IEMA was synthesized according to a previously published protocol [16]. However, this updated formulation features differences in the relative amount of monomers, namely a 40 wt% reduction in HEMA together with an increase of 10-MDP to double the amount in percentage. Relative amounts of each component present in the adhesives are depicted in Table 1. To classify the adhesives according to their pH, each formulation was measured neatly, in triplicate, using a high accuracy pH meter—Crison Basic20 (Crison Instruments, Barcelona, Spain) after pH calibration, at room temperature. The experimental adhesives were stored in opaque sterile bottles, at 4 °C, and kept from light until they were used.

### 2.2. Microtensile Bond Strength (µTBS) to Dentin and Failure Mode Classification

Dentin bond strength of the experimental adhesives and control groups was evaluated by a µTBS test setup, in accordance with the Academy of Dental Material’s µTBS testing guidance [18]. Thirty-six sound permanent human molars (<6 months), extracted due to orthodontic or periodontal reasons, stored at 4 °C in a solution of 1.0% chloramine T, for a maximum of 1 week, approved by the Ethics Committee of Instituto Universitário Egas Moniz (IUEM; Internal Process No. 827), were selected. Mid-coronal dentin was exposed (Accuton 50, Struers A/S, Ballerup, Denmark) and polished (600 grit SiC abrasive paper, Labo-Pol 4, Struers, Ballerup, Denmark); teeth were then randomly assigned to four different materials (FUT, SBU or EM1, EM2) and two different adhesive strategies (etch-and-rinse—ER or self-etch—SE), creating 8 different groups: FUT_ER, FUT_SE, SBU_ER, SBU_SE, EM1_ER, EM1_SE, EM2_ER or EM2_SE. The application mode of each adhesive is portrayed in Table 2.

Following the adhesive procedure, resin build-ups were produced using a Grandio nano-hybrid composite (VOCO, GmbH, Cuxhaven, Germany) A2 shade, in each sample, by applying 2 mm layer increments and light-curing them for 40 s (LED light-curing unit Elipar Deep Cure-S, 3M ESPE, EUA) at zero distance, without the use of an acetate sheet to eliminate the oxygen that inhibits polymerisation at the surface layer. Light intensity output was monitored, showing an average output of 1200 mW/cm^2^, measured every 3 uses with an analog radiometer Demetron 100 (Demetron research Company, Danbury, CT, USA). Samples were further stored for 24 h in distilled water at 37 °C in an incubator (Memmert INE 400, Memmert, Germany). Samples were then sectioned into 1 mm^2^ [±0.2 mm] thick beams, originating microspecimens. The µTBS test was performed using a universal testing machine with a 5 kN loading cell (Shimadzu AG-50kNI SD MS, Shimadzu Corporation, Kyoto, Japan) at a crosshead speed of 0.5 mm/min.

Fractographic analysis was conducted, and fractured beams were classified as adhesive, cohesive (resin or dentin) or mixed failures, with the help of a light stereomicroscope (Olympus/DeTrey, Konstanz, Germany) at 40× magnification.

### 2.3. SEM Analysis of Bonded Interface

For each experimental group, two additional teeth per experimental group stored and prepared under the same conditions as described in Section 2.2, were selected for visualization of the bonding interface using scanning-electron microscopy (SE, FEG-SEM JEOL, model JSM7001F, Tokyo, Japan).

After 24 h in an incubator, the specimens were fixed with glutaraldehyde (2.5%) and 0.1 M of sodium cacodylate, at 4 °C, for 24 h [19]. Subsequently, the specimens were sectioned exposing the adhesive interface with a hard tissue microtome (Accuton 50, Struers A/S, Ballerup, Denmark) and a protocol of demineralization and deproteinization was performed in order to expose the resin tags and remove sectioning debris [20]. Ethanol in ascending concentrations was then used (70%—20 min; 95%—20 min and 100%—30 min, 3×), followed by hexamethyldisilazane (HMDS) in two immersion periods of 10 min each. After 24 h of air-drying, samples were then fixed with a carbon tape and sputter coated with a thin film of gold/palladium on a Q150T ES Turbo-Pumped Sputter coater (Quorum Technologies, Q150T Turbomolecular-pumped coating system, Lewes, UK) with a 15.0 kV electron acceleration beam. The adhesive interface was analyzed using low and high magnifications (500× and 2000×) to visualize the ultrastructure of the hybrid layer.

### 2.4. Nanoleakage

For nanoleakage studies, twenty-four sound human permanent molars were randomly divided between the experimental groups mentioned above and also between two different adhesive strategies, identical to what was described in Section 2.2 (*n* = 3). Sample storage, sectioning, dentin preparation, bonding and the restorative procedure followed what was described under Section 2.2. After preparation, the samples were cut into 0.7 mm thick slices, in a longitudinal direction using a hard tissue microtome (Accutom 50, Struers A/S, Ballerup, Denmark) under water irrigation. Three central slices were selected for each group. All specimens were stored in distilled water and aged for 3 months in an incubator, at 37 °C, 96% moisture and 5% CO_2_. The water was changed every 2 weeks. At the end of this period, the samples were prepared for the observation of nanoleakage by SEM-EDS-SE, FEG-SEM JEOL, model JSM7001F (JEOL, Tokyo, Japan) according to the protocol described by Tay et al., 2002. The specimens were coated in two layers of nail vanish with the exception of 1 mm of the bonding interface which was immersed in an aqueous solution of 50% ammoniacal silver nitrate at 37 °C in darkness for 24 h, rinsed thoroughly in distilled water and immersed in photo developing solution (Sigma Chemical Company, St. Louis, MO, USA) for 8 h under fluorescent light to reduce silver ions into metallic silver grains within voids along the bonded interface [21]. Then the specimens were rinsed again in distilled water for 5 min and the surface vanish was removed.

All samples were prepared for observation using SEM-EDS using the fixation and dehydration procedures previously described in Section 2.3. After HMDS immersion, a polishing procedure using sequential SiC abrasive paper discs was performed. Then, the specimens were sonicated in 100% ethanol for 10 min, demineralized and dried with 0.5% silica-free phosphoric acid for 1 min (Sigma Chemical Company, St. Louis, MO, USA) and finally stored in paper-lined petri dishes. The full extent of the incorporation of silver nitrate along the dentin bonding interface was observed at 200× magnification, aided by Quartz PCI 4.00 software (Quartz Imaging Corporation, Vancouver, BC, Canada).

The silver penetration on the interfacial zone was identified by 2000× magnification micrographs characterized by true intense metallic white spots with globular or reticular shapes. This was also confirmed by a percentage calculation through an energy dispersion spectroscopy (EDS) microanalysis system, coupled to the FEG-SEM (Oxford Inca Energy 250^®^—Oxford Instruments, Oxfordshire, UK). The expression of nanoleakage at the adhesive interface of each sample was calculated by dividing the black area over the total area of the selected zone, in accordance with previously published protocols [22].

### 2.5. Statistical Analysis

All statistical procedures (descriptive and inferential) were performed using statistical software IBM SPSS version 26.0 (IBM Corporation, Armonk, NY, USA). For the analysis of microtensile bond strength (µTBS) data, Linear Mixed Models (LMM) and Generalized Linear Models (GLM) were employed. Measurement of bond strength in different observation units (dentin beams) from the same experimental unit (tooth), may result in non-null correlations and heterogeneous variances that violate the assumptions of GLM, requiring the modeling of these effects through data analysis using LMM. The use of LMM in this analysis requires that the covariance structure between repeated observations in each experimental unit must be known. The selection of the best and simplest covariance structure is made by comparison with the unstructured covariance matrix (UN), the most complex covariance structure that can be adjusted to the data. However, in view of the high number of beams obtained for each tooth, the UN method did not converge to a solution. Thus, prior to the selection of the best covariance structure, a graphical representation of the bond strength values was made, suggesting a homogeneous covariance matrix. In selecting the best structure, the following matrices were then considered for homogeneous variances: 1st-order auto-regressive (AR1), auto-regressive with moving average (ARMA11), composite symmetry (SC), correlated composite symmetry (SCCORREL), Identity (ID) and Toeplitz. The quality of the adjustment obtained with the different matrices was assessed using the information criteria of Akaike (AIC), Hurvich and Tsai (AICC), Bozdogan (CAIC) and Bayes and Schwarz (BIC).

The tooth was considered the statistical unit, for which µTBS means were computed, taking into account the heterogeneity of different beams originating from the same tooth. Similarly, the statistical analysis for the nanoleakage results was carried out using LMM. All tests were conducted at a significance level of 5%.

## 3. Results

### 3.1. Microtensile Bond Strength (µTBS) to Dentin and Failure Modes

According to the information criteria, it appears that the best quality of adjustment of the model, after penalizing the value of −2 Log (Restricted Likelihood), taking into account the number of model parameters, is obtained with a covariance structure of symmetry composed, reflecting the existence of homogeneous variances and also homogeneous correlations. It is important to note that the consideration of heterogeneous covariance structures, such as the diagonal matrix and heterogeneous composite symmetry, resulted in worse quality adjustments, compared to the results shown in Table 3. In view of these results, the model that best translates the effects of the type of adhesive used and the adhesive strategy employed, as well as the interaction between these factors, on immediate bond strength is shown in Table 4.

According to the model, it was possible to confirm that the strategy factor (either ER or SE) showed a statistically significant effect on the µTBS, with significantly higher values being reported with the ER protocol (*p* = 0.026), regardless of the adhesive used. The adhesive choice did not show statistically significant differences, highlighting that in immediate bond strength to dentin, all adhesives performed similarly (*p* = 0.11).

Table 5 portrays µTBS means, minimum, maximum and standard deviation (S.D.) values for the different adhesives and adhesive strategies used.

For each adhesive, the total number of adhesive, cohesive (in composite or dentin) or mixed failures were counted as percentages and they are shown considering the total number observed for both adhesive strategies employed (Table 6). According to the results obtained, the adhesive failure was the most predominant in all groups, followed by cohesive (in composite), in exception to SBU, where cohesive failures in dentin were more prevalent.

### 3.2. SEM Imaging of the Adhesive Interface

SEM micrographs of resin-dentin interfaces of the commercial adhesives (SBU and FUT) and experimental adhesives (EM1 and EM2) show marked differences between the ER and SE adhesive strategies. In general, SEM images presented a thicker hybrid layer (HL) and longer resin tags (RT) when an ER strategy was used, in comparison to SE treated specimens (Figure 1 and Figure 2). This confirms the results obtained in immediate dentin µTBS, where ER showed statistically significant higher values than SE, regardless of the adhesive used.

Among the adhesives studied, although there were no statistically significant differences in µTBS, it is possible to observe some differences in the morphology and ultrastructure of the dentin-resin interface. Following the ER protocol, the SBU commercial group shows a thicker hybrid layer (HL), with a pronounced interaction with the dentinal surface confirmed by the extensive formation of resin tags throughout the bonding area (Figure 1a and Figure 2a). As for FUT, SEM micrographs show a less hybrid layer (HL) compared to others without visible resin tags (Figure 1b and Figure 2b). The experimental adhesives, EM1 and EM2, presented a uniform HL showing good resin penetration within the collagen fibers reinforced with a high density of long resin tags together with substantial ramifications (Figure 1c,d and Figure 2c,d).

Considering the SE adhesive strategy, both commercial adhesives (SBU and FUT) showed a thinner and undefined HL, compared to the ER strategy, with fewer resin tags (Figure 1e,f and Figure 2e,f). Regarding EM1 and EM2, the interface morphology seems to be similar to commercials, although they show a uniform HL (Figure 1g,h and Figure 2g,h).

### 3.3. Nanoleakage

With the data obtained, a mixed model analysis was performed, comparing the mean nanoleakage (measured in µm) in samples obtained from the different adhesive systems under investigation (FUT, SBU, EM1, EM2) applied in two different adhesive strategies (ER, SE). LMM results are shown in Table 7.

It can be seen from Table 7 that a statistically significant interaction was found between the adhesive chosen and the strategy used (*p* < 0.001), highlighting that the nanoleakage observed for a particular type of adhesive depends on the protocol applied. A statistically significant difference between adhesives, for the extent of nanoleakage, was also found (*p* < 0.001). Additionally, there was a statistically significant difference between protocols (*p* = 0.003). Descriptive statistics for nanoleakage data are shown in Table 8 and illustrated in Figure 3, with the planned contrasts shown in Table 9, also.

The following FEG-SEM images (Figure 4 and Figure 5) are representative images of all experimental groups (SBU, FUT, EM1 and EM2) with magnifications of 400× and 2000×, where it is possible to observe nanoleakage of the adhesive interface represented by silver nitrate when the two different adhesive strategies (ER and SE) were investigated, after 3 months aging. In general, it was possible to observe that SBU and FUT when applied in the ER mode revealed a greater amount of silver nitrate, with higher deposition intensity, depicted with a greater accumulation within the HL and immediately underneath it (Figure 4a,b and Figure 5a,b). On the contrary, even under the same strategy, the experimentals EM1 and EM2 showed a lower amount of silver nitrate penetration, thus there was less leakage for these aged samples (Figure 4c,d and Figure 5c,d).

When applied using an SE approach, experimental adhesive EM1 presented a lower intensity of silver nitrate deposits within and underneath the HL (Figure 4g and Figure 5g). Finally, the micrographs also showed that the EM2 applied in SE mode (Figure 4h and Figure 5h) presented less deposition of silver nitrate in the hybrid layer than in ER mode (Figure 4d and Figure 5d).

## 4. Discussion

Universal adhesives (UAs) are the most recent generation of dental adhesives and due to their chemical composition and acidity, they were initially designed to bond with equal effectiveness under both protocols, ER and SE. The operator can then select the best technique according to each specific clinical situation. However, when analyzing the results published in the literature, it appears that the universal adhesives on the market do not yet accomplish these requirements [23,24,25,26].

The adhesive interface obtained with universal adhesives (UAs) is substantially sensible and unstable to hydrolytic processes (which are followed by endogenous enzymatic proteolysis) that can occur immediately after a restorative procedure [10]. The incomplete envelopment of collagen fibrils and virtual spaces during hybridization and the presence of hydrophilic components in UAs make the hybrid layer (HL) highly susceptible to nanoleakage, consequently leading to the failure of dental restorations [12,24]. For this reason, the experimental adhesive formulated in this study aims to overcome this limitation, improving the quality and the cohesive properties of the HL by adding the dendrimer G-IEMA as a cross-linked monomer, instead of Bis-GMA. The results obtained so far revealed that the G-IEMA in dental adhesives improves the degree of conversion and shrinkage, possibly owing to its three-dimensional structure with eight polymerizable methacrylate groups. These characteristics also showed promising results regarding immediate dentin bonding when an ER strategy was performed, however, the chemical composition of this new experimental adhesive may still need optimization to guarantee a good result upon using an SE strategy [16,17].

The results obtained in this study allow partial acceptance of the first null hypothesis given that no statistically significant differences in immediate µTBS results were observed between adhesives, regardless of the strategy. However, the ER protocol presented statistically significant higher µTBS than the SE.

Firstly, it is important to review the pH of the adhesives that were studied. These fit into the intermediately strong (FUT) and ultra-mild (SBU, EM1 and EM2) classification [3], these results are corroborated by similar studies, where the ER strategy of UAs seem to present better results in immediate dentin bond strength [26]. Yet, there is a great disparity in the published results seen in the literature. Some authors argue that there are no statistically significant differences between the ER and SE protocols after 24 h, using universal adhesives [23,27], while others report a better performance with the ER approach [28]. These differences can be attributed to the chemical composition of each adhesive, and it is agreed that the performance of universal adhesives is thus very material-dependent ([29]. According to recent meta-analytical data shown in Rosa et al. (2015) [30] and Cuevas-Suárez et al. (2019) [26], the performance of UAs is dependent on the pH of the adhesive, the bonding substrate (enamel or dentin) and the strategy used. Cuevas-Suárez et al. (2019) also pointed out that the stability of UAs depends largely on the pH, and that when bonding to dentin is desired, mild universal adhesives (pH ~ 2) seem to outperform the rest regarding bond integrity and stability, in both strategies. The pH of these formulations was lower than that of their antecessors [16,17] owing to the increase of phosphate groups per unit volume of adhesive, since the 10-MDP wt% was increased in the present formulations. Compared to previous results, this maintained the immediate bond strength of the adhesive containing G-IEMA, while it lowered the µTBS in EM1. This may be attributed to the decrease of HEMA, known to facilitate diffusion during hybridization which could have affected EM1. Interestingly, in EM2, the presence of G-IEMA may be compensating for the decrease of HEMA since µTBS results were similar.

Despite this, it should be noted that there were no statistically significant differences in µTBS values of different adhesives. The fact is that the chemical composition of the tested adhesives is similar and the small differences that do exist are most probably not relevant enough to translate into a statistically significant difference in immediate bond strength. It is likely that those differences may be important in long term adhesion after an aging period. Such studies are underway and will be addressed in the future.

Bond strength results were complemented with an SEM analysis of the ultrastructure of the resulting adhesive interface. When the SEM micrographs of the present study were analyzed, it was clear that the ER strategy produced a much thicker HL, allowing more resin impregnation within the collagen network and dentinal tubules, when compared to SE. These differences observed in the SEM corroborate the bond strength results. Furthermore, this is widely supported in the literature as acid-etched dentin results in removal of the smear layer, smear plugs withing tubules and more collagen exposure up to a considerable depth of 5 µm (thus thicker HL) [31,32,33]. After applying the adhesive and carrying out polymerization in situ, a thick HL is then formed which, together with the resin tags inside the dentinal tubules, form a micromechanical retentive mechanism contributing to the overall bond strength [31]. Comparing the SEM images with the results published by other authors, there is a similar dentin hybridization with the UAs that were tested [10,34]. Wagner et al. (2014) also showed a better dentin penetration with the ER approach than with the SE approach for the Futurabond Universal, Scotchbond Universal Adhesive and All-Bond Universal. The effective penetration of the UAs into dentin was also proved by evidence of long resin tags.

Additionally, the experimental adhesives EM1 and EM2 seem to have a more cohesive and richer HL as well as a greater number of resin tags, reaching higher depths, with formation of lateral tags, contrasting to the interface of commercials SBU and FUT. However, these differences did not translate into better bond strength, as tags are known to be an additional supplement that gives extra mechanical retention, not directly contributing to improve bonding [35]. The quality of the HL is the most important factor for the stability of the adhesive interface and consequently the longevity of tooth restoration [3,36,37]. Even so, higher density of resin tags could mean that the experimental UAs are interacting more with the surface of dentin. In fact, G-IEMA in EM2 could be playing a role in pulling the adhesive into the collagen network, due to the formation of hydrophobic interactions between the monomer-collagen, or interactions via the -NH groups. In this group, a higher number of resin tags per surface area were seen.

With aging taking place in the oral environment, mechanical, thermal and chemical processes breakdown the HL, making it more susceptible to bond deterioration and bacterial infiltration [12,36]. Cyclic and inter-dependent mechanisms of hydrolytic and enzymatic degradation are today considered the chief culprits of this breakdown [37]. However, the chemical composition of the restorative polymers, namely the type and relative percentage of solvents and the ratios of hydrophilic and acidic functional monomers in adhesives, regulate their affinity towards water [25,26]. In consequence, since UAs are presented in a single solution with all the components mixed, they act as semi-permeable membranes due to the existing disparities between the hydrophilic and hydrophobic components. Consequently, this increase in water content in the HL is associated with monomer dilution, adding difficulty to their approximation during polymerization and fostering of water ingress, which is bound to increase with time [38,39,40]. Moreover, the aqueous and acidic environment also activates the MMPs responsible for the enzymatic degradation of collagen [41,42]. These factors described are responsible for the development of nanoleakage phenomena.

According to some authors, the nanoleakage process starts as early as within 24 h after restoration placement. It can be considered that it begins from the moment the collagen is left unprotected [12,31,43].

Dental adhesives which are more acidic and have higher water content and/or hydrophilic monomers in their composition, logically, have a greater chance of nanoleakage [14]. On the contrary, mild and ultra-mild adhesives that contain functional monomers seem to be more stable. This may be due to a couple of reasons. Firstly, these systems demineralize less, exposing less collagen, thus diminishing the chances of matrix metalloproteinase (MMP) action [43,44]. Secondly, the calcium partially dissolved owing to hydroxyapatite-depletion in dentin, is kept in a stable form to allow chemical bonding to functional monomers, such as 10-MDP [32,45,46]. This reinforces the micromechanical retention and the cohesive properties of the bonding interface. As already corroborated by a considerable number of studies, UAs that contain 10-MDP are considered by several authors to be more effective in long term dentin adhesion when applied with SE rather than ER [12,14]. Lowering the pH by increasing the 10-MDP content, as in these formulations, is expected to contribute to further chemical bonding mechanisms which may not directly translate into higher immediate bond strengths but could provide additional long-term stability. Regarding the formulation containing G-IEMA, previous results have shown that water sorption and solubility data are comparable to the commercial formulations assessed in this study, indicating that aging properties of this experimental formulation may be similar, although this warrants future study [16].

Concerning nanoleakage evaluation in this study, the results supported the rejection of the second null hypothesis, since there were statistically significant differences between adhesives and strategies.

After 3 months aging, the SBU and FUT presented statistically significant higher nanoleakage expression, under an ER strategy in comparison to that of the SE strategy. Marchesi et al. (2014) [47] and Luque-Martinez et al. (2014) [48] also tested the SBU, reporting similar results. Yet when experimentals EM1 and EM2 were considered, it was possible to observe the opposite, presenting significatively less nanoleakage using the ER strategy.

Some factors may possibly justify these results. On the one hand, the experimental adhesives are formulations with lower viscosities than the commercials, which may facilitate better impregnation of the adhesives upon hybridization of the collagen fibrils, closing and filling in virtual spaces [43]. This was assessed visually and also confirmed with previous data which also relates to their different densities [16]. On the other hand, a lower content in hydrophilic monomers of these new formulations, such as HEMA, most probably made the adhesive interface less permeable to water. HEMA is a hydrophilic monomer used as a co-solvent that improves bonding to intrinsically moist dentin, although it is easily degradable as it attracts water. Moreover, it competes with 10-MDP for linkages to hydroxyapatite, harming the stability of the bonding interface mainly in the SE strategy [49,50]. Finally, the type of solvents may also be another differentiating factor, since there are no certainties in the exact solvent ratio of commercial materials [51]. The concentration of ethanol may be lower in commercial adhesive systems than in experimental systems and could in turn hamper solvent evaporation, as previously described by several authors, thereby increasing water retention at the adhesive interface [52].

Considering the results of nanoleakage with the SE strategy, EM1 showed statistically significantly less silver nitrate deposition compared to alternatives. Since the EM1 and EM2 have exactly the same chemical composition differing only in the main monomer, where EM1 contains Bis-GMA and EM2 contains G-IEMA in its replacement, the reason for these differences could possibly be attributed to the different pH of adhesives.

Proper encapsulation of the smear layer by EM1, could have played a role in its ability to better infiltrate the substrate during the formation of the HL, whereas EM2 had a more limited interaction over the surface area. According to Van Meerbeek et al., (2020), the EM1 can be classified as a mild adhesive (pH = 2.02) and EM2 as an ultra-mild adhesive (pH = 2.24). The ultra-mild adhesives present an added challenge in dentin penetration when used with the SE technique [53]. The application time of a UA is directly related to its etching efficacy and smear layer dissolution, thus improving infiltration and bond strength [54]. Thus, adjusting the pH in future formulations could disclose the specific role of G-IEMA in hybridizing the dentinal substrates.

Considering all the results obtained thus far, further support of the use of dendrimers in dental adhesive formulations was obtained. A BPA-free adhesive containing G-IEMA is not only a viable approach but comparable to existing commercial adhesives in some properties, while showing better results in others. However, further research is required to better understand the usefulness of these dendrimers when hybridizing dentin.

## 5. Conclusions

The present study successfully developed an improved and novel formulation of an experimental adhesive containing a second-generation dendrimer as a bulk monomer, in replacement of Bis-GMA. Overall, based on the laboratory study conducted and considering its limitations, it can be concluded that immediate bond strength to dentin was similar for all adhesives tested. However, with EM1 and EM2, the adhesive interface was richer, with a higher density of long resin tags which could be attributed to differences in chemistry and viscosity of the experimental adhesives.

After 3 months aging, EM1 and EM2 continued to prove better results, revealing statistically significantly less nanoleakage expression under the ER strategy than commercials. With the SE protocol, although EM1 showed the lowest nanoleakage, there were no differences between EM2 and commercials, meaning that with this protocol these adhesives showed similar behavior after aging. Low viscosity, a low content of hydrophilic monomers content and higher relative percentage of volatile solvents and functional monomers are the possible factors attributed to the differences observed between commercial and experimental adhesives.

The present research supports the use of dendrimers instead of classical linear monomers, upon formulation of new dental adhesives, and adds to previous research conducted with adhesives containing G-IEMA. This opens a new path of research and development of next-generation dental materials containing dendrimers.

## 6. Patents

This work resulted in a national patente, registered under the No. 115,064—Formulação para um sistema adesivo dentário universal, contendo um monómero de reticulação dedrítico de segunda geração Joana Vasconcelos e Cruz, Luísa Gonçalves and Mário Polido. (2019). Egas Moniz Cooperativa de Ensino Superior, C.R.L.

## Figures and Tables

**Figure 1 polymers-14-01462-f001:**
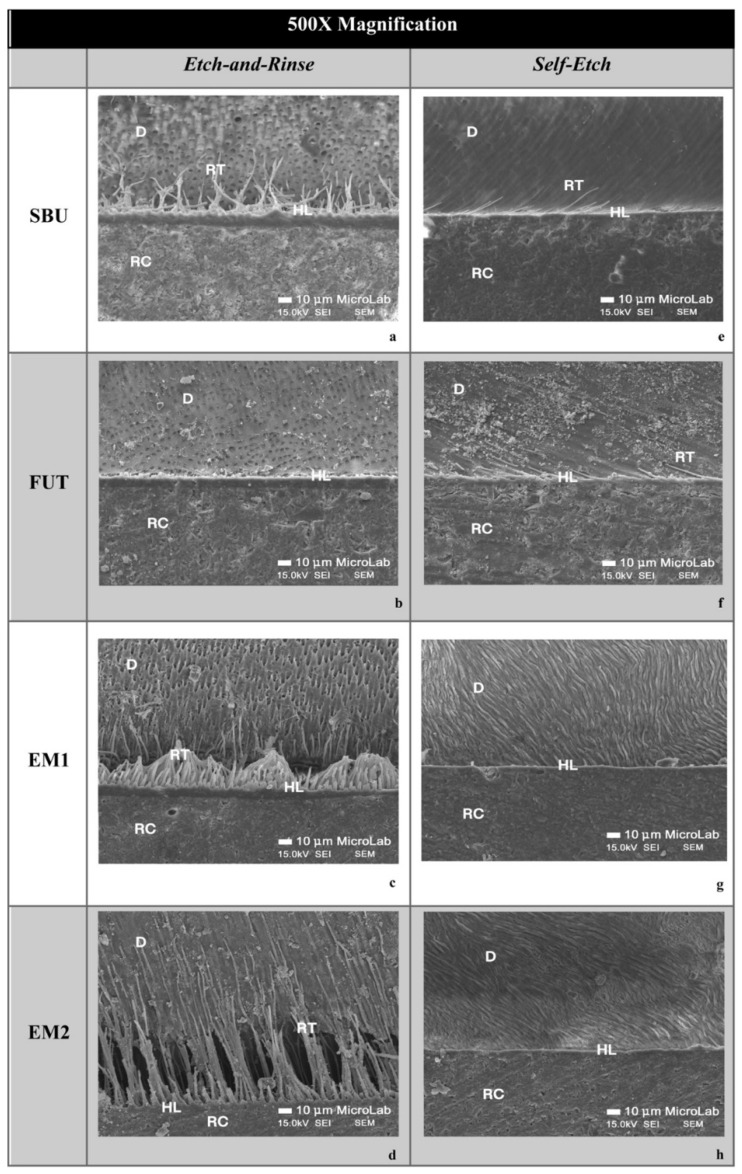
FEG-SEM micrographs of the adhesive interface at 500× magnification. (**a**–**d**) adhesives using an ER strategy; (**e**–**h**) SE strategy. It is possible to observe the resin tags (RT), resin composite (RC), dentin (D) and hybrid layer (HL). ER strategies show thicker HLs, while SE is shown to be thinner and non-uniform in the commercials. EM1 and EM2 show a great disparity in resin tag length and density compared to SBU and FUT.

**Figure 2 polymers-14-01462-f002:**
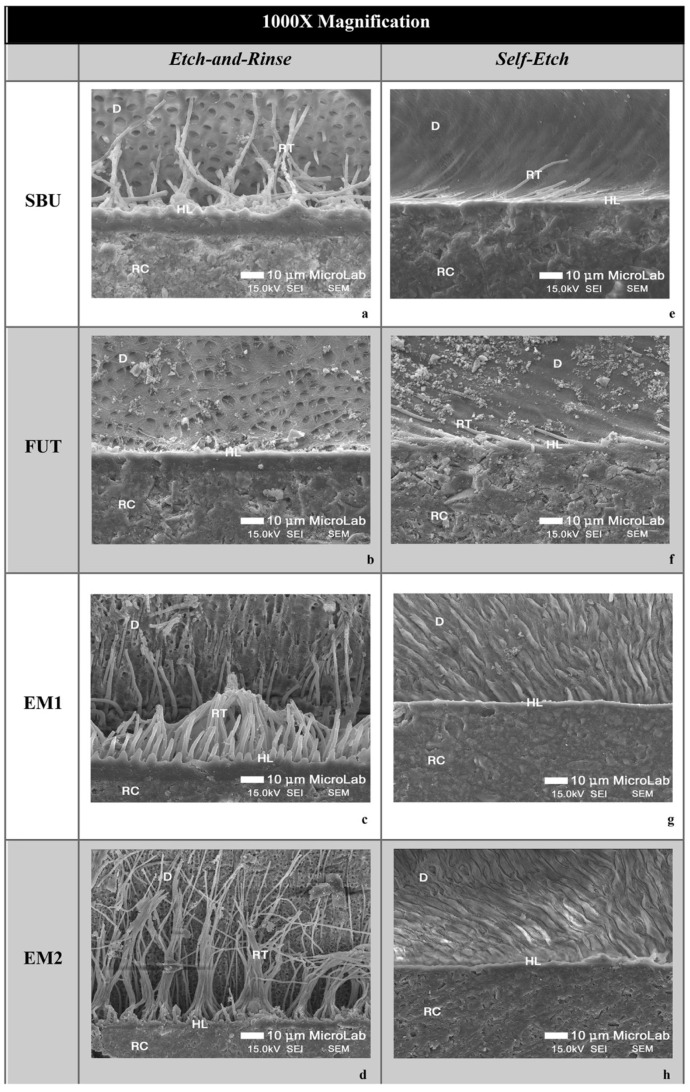
FEG-SEM micrographs of the adhesive interface at 1000× magnification. (**a**–**d**) adhesives using an ER strategy; (**e**–**h**) SE strategy. Resin tags are represented as (RT), resin composite (RC), dentin (D) and hybrid layer (HL). In a higher magnification it is possible to observe, again, long, dense and thick resin tags with EM1 and EM2, although EM2 seems to show better tubule penetration in opposition to EM1.

**Figure 3 polymers-14-01462-f003:**
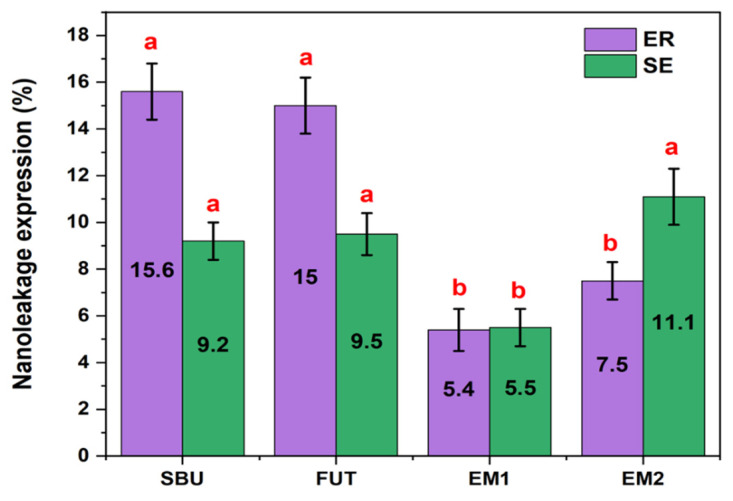
Graphical representation of nanoleakage, by mean percentage scores, in all experimental groups under study, depending on the adhesive strategy used. Different letters (a,b) refer to statistically significant differences among adhesives and protocols.

**Figure 4 polymers-14-01462-f004:**
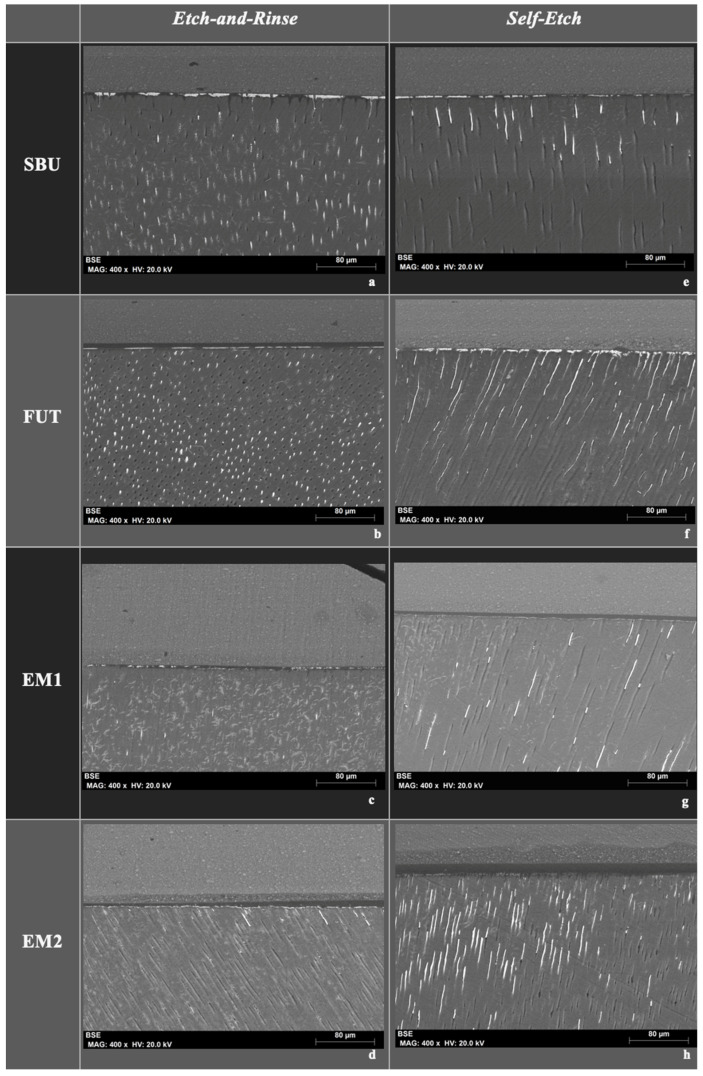
FEG-SEM images with backscattered electron detector of all universal adhesive systems (experimental and commercial) when applied under ER or SE strategies, at 400× magnification. (**a**) shows SBU in etch-and-rinse, while (**e**) shows SBU in self-etch mode, (**b**) shows FUT in etch-and-rinse and (**f**) in self-etch, (**c**) shows experimental EM1 in etch-and-rinse and (**g**) in self-etch and (**d**) EM2 in etch-and-rinse while (**h**) in self-etch mode. It is possible to observe the silver nitrate deposit zones on the adhesive interface which correspond to white spots with a granular appearance. These were more intense in ER strategies, compared to SE. Experimental materials EM1 and EM2 show a competitive nanoleakage performance to commercials, with EM1 showing less intensity than alternatives.

**Figure 5 polymers-14-01462-f005:**
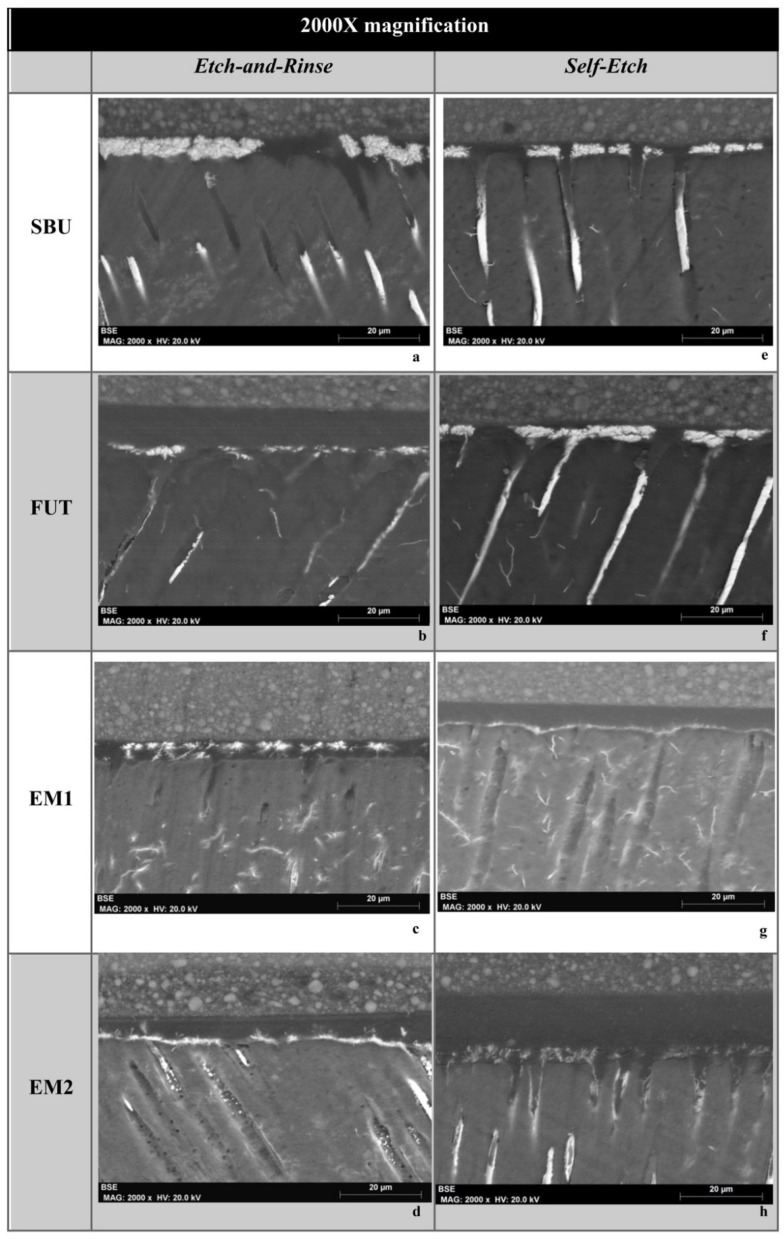
FEG-SEM images of the ER/SE adhesive interfaces, at 2000× magnification. (**a**) shows SBU in etch-and-rinse, while (**e**) shows SBU in self-etch mode, (**b**) shows FUT in etch-and-rinse and (**f**) in self-etch, (**c**) shows experimental EM1 in etch-and-rinse and (**g**) in self-etch and (**d**) EM2 in etch-and-rinse while (**h**) in self-etch mode. Silver nitrate deposit zones on the adhesive interface correspond to white spots with a granular appearance. Similarly, to the image above, it is possible to discern that ER showed higher nanoleakage expression than SE strategies. Experimentals EM1 and EM2 show less pronounced nanoleakage under SE compared to commercials.

**Table 1 polymers-14-01462-t001:** (**A**). Relative percentage of components according to information derived from manufacturers’ and safety datasheets. EM1/EM2 formulations were also taken from Vasconcelos e Cruz et al. (2020), which shows the relative percentage of monomers (wt%), while Table B shows the photoinitiator system and the solvents in wt%. (**B**). Table shows the wt% of the components in the photoinitiator system and the solvents present in adhesives.

**A**		**Monomers (wt%)**					
**Adhesive**	**pH**	**Bis-GMA**	**UDMA**	**HEMA**	**G-IEMA**	**TEGDMA**	**10-MDP**
SBU	2.7	15–25	-	15–25	-	-	N/A
FUT	1.7	10–25	2.5–5	10–25	-	-	N/A
EM1	2.0	10–15	5–10	10–25	0	5–10	5–10
EM2	2.2	-	5–10	10–25	10–15	5–10	5–10
**B**		**Photoinitiator system and solvents (wt%)**					
**Adhesive**	**pH**	**CQ**	**BDA**	**HPDPI**	**Ethanol**	**Water**	
SBU	2.7	<2	<2	-	15–25	10–15	
FUT	1.7	<2.5	0	-	10–25	-	
EM1	2.0	<2	<2	1.5–2	18–28	10–15	
EM2	2.2	<2	<2	1.5–2	18.28	10–15	

10-MDP—10-methacryloyloxy decyldihydrogen phosphate; BDA—ethyl4-dimethylaminebenzoate; Bis-GMA—Bisphenol A-glycidyl methacrylate; CQ—camphoroquinone; G-IEMA—G(2)-isocyanatoethyl methacrylate; HEMA—2-hydroxyethyl methacrylate; HPDPI—diphenyliodoniumhexafluorophosphate; TEGDMA—triethylene glycol dimethacrylate; UDMA—urethane dimethacrylate.

**Table 2 polymers-14-01462-t002:** Description of the application mode for each adhesive and respective adhesive strategy (ER/SE).

Group	Protocol	Application Mode
**SBU** **FUT**	**ER**	Etching with orthophosphoric acid (15 s; Kerr Etchant Gel 37.5%, Kerr, Orange, CA, USA) Water rinsing and air drying (15 s; 5 s) Spreading and brushing the adhesive for 20 s Evaporating solvents for 5 s Light-curing for 10 s *
	**SE**	Spreading and brushing the adhesive for 20 s Evaporating solvents for 5 s Light-curing for 10 s *
**EM1** **(with Bis-GMA)** **EM 2** **(with G-IEMA)**	**ER**	Etching with orthophosphoric acid (15 s; Kerr Etchant Gel 37.5%, Kerr, Orange, CA, USA) Water rinsing and air drying (15 s; 5 s) Spreading and brushing the adhesive for 20 s Evaporating the solvents for 5 s Light-curing for 60 s *
	**SE**	Spreading and brushing the adhesive for 20 s Evaporating the solvents for 5 s Light-curing for 60 s *

* Light-curing parameters were identical to the description found under Section 2.2.

**Table 3 polymers-14-01462-t003:** Information on the different covariance structures.

	AR1	ARMA11	CS	CSCORREL	ID	TOEPLITZ
−2 Restricted Log Likelihood	4814.08	4801.72	4801.71	4801.72	4814.15	NC
Akaike’s Information Criterion (AIC)	4818.08	4807.72	4805.72	4805.72	4816.15	NC
Hurvich and Tsai’s Criterion (AICC)	4818.09	4807.76	4805.74	4805.74	4816.16	NC
Bozdogan’s Criterion (CAIC)	4828.84	4823.86	4816.48	4816.48	4821.53	NC
Schwarz’s Bayesian Criterion (BIC)	4826.84	4820.86	4814.48	4814.48	4820.53	NC

**Table 4 polymers-14-01462-t004:** Tests for type III fixed effects (LMM) of µTBS data.

Source	Num. Df	Den. Df	F	Sig.
Intercept	1	28.30	1049.62	<0.001
Adhesive	3	28.05	2.17	0.114
**Strategy**	**1**	**28.31**	**5.51**	**0.026**
Adhesive * Protocol	3	28.05	1.43	0.255

**Table 5 polymers-14-01462-t005:** Descriptive statistics for the different experimental groups, commercial and experimental, according to the adhesive strategy (ER/SE), S.D. is standard deviation.

Adhesives	Strategy	µTBS (MPa)			
		Min	Max	Mean	S.D.
SBU	ER	17.6	35.5	**26.2**	**6.5**
FUT		30.2	35.9	**33.9**	**3.2**
EM1		26.3	42.6	**32.4**	**6.4**
EM2		21.3	33.9	**29.3**	**5.3**
SBU	SE	19.5	28.1	**23.7**	**3.1**
FUT		21.2	26.2	**24.1**	**2.6**
EM1		18.8	34.4	**26.8**	**6.5**
EM2		26.0	33.7	**30.2**	**3.4**

**Table 6 polymers-14-01462-t006:** Summary of the fractographic analysis showing the different types of failures seen in each experimental group (presented as %).

Failures (%)	Adhesive	Cohesive Dentin	Cohesive Composite	Mixed
SBU	72.3	23.2	4.5	0
FUT	84.5	3.6	10.0	1.8
EM1	72.5	10.4	13.0	4.1
EM2	77.5	6.9	12.4	3.2

**Table 7 polymers-14-01462-t007:** Tests for type III fixed effects (LMM) of nanoleakage data.

Source	df	Mean Square	F	Significance
Intercept	1	35463.04	788.36	0.000
Strategy	1	393.98	8.76	0.003
Adhesive	3	1113.31	24.75	0.000
Strategy * Adhesive	3	496.53	11.04	0.000

**Table 8 polymers-14-01462-t008:** Descriptive statistics for the mean results of nanoleakage (in %) for the different experimental groups, according to their adhesive strategy or application mode (ER/SE). CI represents confidence interval.

Adhesive	Strategy	Mean	S.D.	95% CI	
				Min.	Max.
SBU	ER	15.6	1.2	13.3	17.9
	SE	9.2	0.8	7.5	10.9
FUT	ER	15.0	1.2	12.6	17.4
	SE	9.5	0.9	7.7	11.2
EM1	ER	5.4	0.9	3.6	7.2
	SE	5.5	0.8	3,9	7.0
EM2	ER	7.5	0.8	5.9	9.1
	SE	11.1	1.2	8.9	13.4

**Table 9 polymers-14-01462-t009:** Planned contrast results for all variables, indicating statistically significant differences.

Adhesive	Protocol	*p* *
FUT vs. SBU	ER	n.s.
FUT vs. SBU	SE	n.s.
FUT vs. EM1	ER	<0.001
FUT vs. EM1	SE	0.001
FUT vs. EM2	ER	<0.001
FUT vs. EM2	SE	n.s.
SBU vs. EM1	ER	<0.001
SBU vs. EM1	SE	0.002
SBU vs. EM2	ER	<0.001
SBU vs. EM2	SE	n.s.
EM1 vs. EM2	ER	n.s.
EM1 vs. EM2	SE	<0.001

* a statistically significant difference should be noted whenever *p* < 0.05.

## Data Availability

Data may be available upon request from the corresponding author.

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
