# Peer review of "Improving Properties of an Experimental Universal Adhesive by Adding a Multifunctional Dendrimer (G-IEMA): Bond Strength and Nanoleakage Evaluation"

_polymers, 2022, doi:10.3390/polym14071462_

Round 1

Reviewer 1 Report

In this manuscript, the authors reported the re-formulation of a dental adhesive containing G-IMEA, a G-2 dendrimer containing eight methacrylate groups, to overcome the limitations of the previous reported formulation. Compared with commercial adhesive, the resulting adhesive has comparable bonding strength to dentin and improved nano-leakage performance. All the methods are clearly described. The results are adequately presented. However, the difference of this formulation and its performance compared to the previously reported formulations with G-IMEA are not discussed, which might be the most significant point of this manuscript. In summary, I will suggest its publication in Polymers after a major revision.

To help visualize the improvement of this work, can the authors present and discuss the differences of this formulation and performance with the previous formulations? For example, how the PH value of this formulation improves its efficacy of self-etch approach? What about their aging properties?

Author Response

We thank the reviewer for his pertinent comments and suggestions, which greatly improved the scientific quality of the present manuscript, and we provide a point-by-point response for him below.

Point 1: To help visualize the improvement of this work, can the authors present and discuss the differences of this formulation and performance with the previous formulations?

Reply: In order to clarify major differences in between formulations, the authors re-wrote the Methods Section (pag.3) from line 114 to 116, which now reads “However, this updated formulation features main differences changes in the relative amount of monomers, namely a 40 wt% reduction in HEMA together with an increase of 10-MDP to double the amount in percentage.” A new paragraph was also inserted in the discussion section from lines 422 to 428, which highlights the importance of the concentration of monomers on the global performance of the adhesives; it now reads “(…) The pH of these formulations was lower than that of their antecessors [16, 17] owing to the increase of phosphate groups per unit volume of adhesive since the 10-MDP wt% was increased in the present formulations. Compared to previous results, this maintained the immediate bond strength of the adhesive containing G-IEMA, while it lowered the µTBS in EM1. This may be attributed to the decrease of HEMA percentage, known to facilitate diffusion during hybridization which could have affected EM1. Interestingly, in EM2, the presence of G-IEMA may be compensating for the decrease of HEMA since µTBS results were similar.(…) “.

Point 2: For example, how the pH value of this formulation improves its efficacy of self-etch approach? What about their aging properties?

Reply: We acknowledge the question raised by the reviewer. As mentioned above, to address this question clearly, the authors inserted a new paragraph to the Discussion section (pag.#) from lines 422 to 428. Regarding aging properties, which were not assessed for bond strength outcomes, a short sentence was also inserted to the text (line 497 to 502) to clarify the point brought up by the reviewer.

Reviewer 2 Report

This is an interesting study on an important topic, which is on a path of improving the properties of contemporary dental adhesive systems. The script is generally very well-written and the study is well-performed. I have only one important methodological remark to be addressed by the authors.

In the manuscript, you stressed multiple times that the experimental G-IEMA-modified adhesives are expected to discourage the degradation of the hybrid layer, leading to improved longevity of the bonded interface. Also, the Introduction section states that while there is some data for immediate bond strength, the behavior of experimental adhesives regarding bond strength after an aging period remains unknown. Given that bond strength usually deteriorates due to the hybrid layer degradation which your experimental composites are intended to prevent, why did the present study evaluate the immediate bond strength? Introducing a simple artificial aging protocol such as storage in distilled water at the body temperature would allow answering the question of whether the experimental adhesives are really able to improve the stability of the adhesive layer compared to their commercial counterparts and Bis-GMA-containing adhesives. It is known that most contemporary adhesive systems perform exceptionally well immediately after bonding, only to show significant differences in performance after being subjected to hydrolytic degradation/aging. This is why the “aged” bond strength is much more valuable and discriminative parameter than the immediate bond strength.

The Discussion section also highlights the importance of artificial aging for discerning more subtle differences among the adhesives: “The fact is that the chemical composition of the tested adhesives is similar and the small differences that do exist are most probably not relevant enough to translate into a significant difference in immediate bond strength. It is likely that those differences may be important in long term adhesion after an aging period.” This adds to my previos question: Why not simply introduce an aging period and be able to identify these differences?

Also, you mentioned in the Discussion section that nanoleakage begins within 24 h after restoration placement. Why was then nanoleakage evaluated 3 months after the restoration placement, while bond strength was tested immediately (24 h) after restoration placement? Such a discrepancy between aging periods appears illogical and should be clarified. Why wait for 3 months to evaluate a property that is expected to occur shortly after specimen preparation (nanoleakage), and evaluate bond strength immediately after specimen preparation, although it is well-known that bond strength needs some time to demonstrate more “clinically relevant” behavior due to hybrid layer degradation?

Please also note some minor remarks:

In the Materials and Methods section, please specify the resin composite used for making the build-ups on top of the adhesive layer. There is usually some interaction between the resin composite and adhesive (primarily due to the elastic modulus of the composite), hence it is important to report with sufficient precision the characteristics of the composite material used for build-ups.

Why do you sometimes refer to statistically significant differences as “significative differences”?

Figure 3: Please specify the meaning of the letters (a, b) above the bars, which are presumably designations of statistically homogeneous groups. Please also clarify at which level were these comparisons performed (it seems that the comparisons were performed among adhesives, within the SE group and the ER group but this should be explicitly specified).

Author Response

The authors would like to express their thankfulness towards the reviewer for raising valid points and enhancing the scientific discussion which greatly improved the quality of the present manuscript. We provide a point-by-point response for him below.

Point 1: In the manuscript, you stressed multiple times that the experimental G-IEMA-modified adhesives are expected to discourage the degradation of the hybrid layer, leading to improved longevity of the bonded interface. Also, the Introduction section states that while there is some data for immediate bond strength, the behavior of experimental adhesives regarding bond strength after an aging period remains unknown. Given that bond strength usually deteriorates due to the hybrid layer degradation which your experimental composites are intended to prevent, why did the present study evaluate the immediate bond strength?

Reply: Firstly, we would like to thank the reviewer for the points raised and changes requested, as these have substantially improved the scientific quality of the manuscript. In the present study we decided to evaluate immediate bond strength as this study features an updated formulation to a previously published paper by our group. Immediate bond strength serves as a direct screening laboratory test to rapidly compare experimental formulations to commercials – as was the case in the present study. This was done in line with previous work undertaken by our group, and a comparison to these results can be seen in the Discussion section, now revised (lines 430-436 and 497-502). We have also justified the differences between these new formulations and their antecessors in the Methods section (line 114-116).

Point 2: The Discussion section also highlights the importance of artificial aging for discerning more subtle differences among the adhesives: “The fact is that the chemical composition of the tested adhesives is similar and the small differences that do exist are most probably not relevant enough to translate into a significant difference in immediate bond strength. It is likely that those differences may be important in long term adhesion after an aging period.” This adds to my previous question: Why not simply introduce an aging period and be able to identify these differences?

Reply: The reviewer raises a valid point and we thank him for his question and suggestion. In fact, we have introduced a sentence in the Discussion addressing this point (please see lines 441-442). The aim of this present research was to assess immediate bond strength and nanoleakage as a rapid screening of the experimentals, relating to previously published papers, which serves as a starting point for future studies. As added to the discussion, assays featuring an aging period are underway and such data will be published in the future. Despite this, the present data featured in this paper was accomplished using sound methodology and we believe it is certainly worth publishing, as an initial exploratory study for the updated formulations.

Point 3 Also, you mentioned in the Discussion section that nanoleakage begins within 24 h after restoration placement. Why was then nanoleakage evaluated 3 months after the restoration placement, while bond strength was tested immediately (24 h) after restoration placement? Such a discrepancy between aging periods appears illogical and should be clarified. Why wait for 3 months to evaluate a property that is expected to occur shortly after specimen preparation (nanoleakage), and evaluate bond strength immediately after specimen preparation, although it is well-known that bond strength needs some time to demonstrate more “clinically relevant” behavior due to hybrid layer degradation?

Reply: Again, we thank the reviewer for this comment. We have partially covered this question in the replies written above (in the previous points). The reviewer is right in mentioning this, and we would like for him to know that long-term bond strength data is underway for future publications. Immediate bond strength data is extremely important as it is the first set of data that determines the viability of an adhesive formulation – which is why it was planned for this study. Regarding nanoleakage, we opted for a laboratory long-term evaluation, as 3 months is considered relevant and comparable to similar literature that also assessed nanoleakage, whereas immediate testing of leakage is seldom found and not as accepted scientifically.

Point 4: In the Materials and Methods section, please specify the resin composite used for making the build-ups on top of the adhesive layer. 

Reply: We absolutely agree that the indication of the commercial resin used in the build-ups is essential for the correct interpretation of the data. By mistake, the resin trademark was not specified in the Materials and Methods section. The authors corrected the text on page 4, line 165/166, to “Following the adhesive procedure, resin build-ups were made using a Grandio nano-hybrid composite (VOCO, GmbH, Cuxhaven, Germany) (…).”

Point 5: Why do you sometimes refer to statistically significant differences as “significative differences”?

Reply: The authors agree with the suggestion and have uniformized all expressions throughout the document by changing them to “statistically significant differences”.

Point 6: Figure 3: Please specify the meaning of the letters (a, b) above the bars, which are presumably designations of statistically homogeneous groups. Please also clarify at which level were these comparisons performed (it seems that the comparisons were performed among adhesives, within the SE group and the ER group but this should be explicitly specified).

Reply: The letters (a, b) identify the statistical differences identified between adhesives and protocols. For better clarification for readers, the authors added the following sentence to the legend of Figure 3 (page 12 and line 360) “Different letters (a.b) means statistically significant differences among adhesives and protocols.”

Reviewer 3 Report

The authors have reported the addition of dendrimers into dental materials. The results are presented nicely and explained well. I recommend its publication after minor revision. 

  1. Abstract: Too many abbreviations are there, it's confusing to read. Please modify.
  2. Inside the manuscript too, many abbreviations are not explained at first instance. Please check the whole manuscript. 
  3. In figure 5b, the image is flipped. Please check and change it. 
  4. In Figures 4 and 5, sections a,b c, etc are not addressed in the caption. Please add details.
  5. whether the authors carried out any experiments to compare the viscosity? because the conclusion and results contain the terminology a lot. 
  6. tables look too busy. please rearrange or modify to make it readable. 
  7. Minor comment: Please use only two numbers after decimal points while representing the data. So that it will be easy for readers. 

Author Response

Point 1: Abstract: Too many abbreviations are there, it's confusing to read. Please modify.

Reply: We acknowledge the comment made by the referee. In fact, the abstract presented many abbreviations. To simplify the reading, the abstract was fully revised, and changes can be seen throughout.

Point 2: Inside the manuscript too, many abbreviations are not explained at first instance. Please check the whole manuscript. 

Reply: As the reviewer suggested, we revised all the manuscript and introduced the meaning of the abbreviations. Please see changes highlighted throughout the entire manuscript. We believe that reading has now been simplified with these changes.

Point 3: In figure 5b, the image is flipped. Please check and change it. 

Reply: We thank the reviewer for bringing this up as we had not noticed such detail. This image has been replaced with another image with the proper orientation.

Point 4: In Figures 4 and 5, sections a,b c, etc are not addressed in the caption. Please add details.

Reply: We thank the reviewer for this pertinent comment, and we have now added the information belonging to a), b), c) etc. in the figure captions for Figure 4 and 5.

Point 5: whether the authors carried out any experiments to compare the viscosity? because the conclusion and results contain the terminology a lot. 

Reply: We acknowledge this information and comment and we have now clarified this in the discussion section (please see lines 514-515). This was assessed visually but also confirmed with previous data that evaluated the same adhesives.

Point 6: tables look too busy. please rearrange or modify to make it readable. 

Reply: We agree with the reviewer on this point raised and we have now changed Tables 1 and 2 to make them more readable and easier to follow. Table 1 has been split into two separate component tables and we believe it enhanced its readability.

Point 7: Minor comment: Please use only two numbers after decimal points while representing the data. So that it will be easy for readers. 

Reply: We agree and thank the suggestion. To simplify the reading the authors corrected the tables 3, 4 and 7 using onlu two decimal numbers.

Round 2

Reviewer 1 Report

All my questions have been addressed. I will suggest its publication in present form.

Author Response

The authors would like to thank the reviewer once again.

Reviewer 2 Report

Thank you for revising the manuscript. I have no further remarks.

Author Response

(The authors gave the same response as above.)
